

# Modelling the effect of condensed-phase diffusion on the homogeneous nucleation of ice in supercooled water

Kathryn Fowler[1], Paul Connolly[1], and David Topping[1]

[1]Centre for Atmospheric Science, The School of Earth and Environmental Sciences, The University of Manchester

**Correspondence:** Paul Connolly (paul.connolly@manchester.ac.uk)

**Abstract.**

In-situ studies of low temperature cirrus clouds have found unexpectedly low ice crystal numbers and consistently high supersaturations, which suggest that our understanding of the freezing mechanisms under these conditions are incomplete. Computational models typically use homogeneous nucleation to predict the ice nucleated in supercooled water. However, the existence of ultra-viscous organic aerosol in the upper troposphere has offered alternative ice nucleation pathways, which have been observed in laboratory studies. The possible effects of aerosol viscosity on cloud micro-physical properties have traditionally been interpreted from simple model simulations of an individual aerosol particle based on equilibration timescales. In this study, to gain insight into the formation of ice in low temperature cirrus clouds, we have developed the first cloud parcel model with bin micro-physics to simulate condensed phase diffusion through each individual aerosol particle. Our findings demonstrate, for the first time, the complex relationship between the rate of ice formation and the viscosity of secondary organic aerosol, driven by two competing effects - which cannot be explained using existing modelling approaches. The first is inhibition of homogeneous ice nucleation below 200K, due to restricted particle growth and low water volume. The second occurs at temperatures between 200K and 220K, where water molecules are slightly more mobile and a layer of water condenses on the outside of the particle, causing an increase in the number of frozen aerosol particles. Our new model provides a basis to better understand and simulate cirrus cloud formation on a larger scale, addressing a major source of uncertainty in climate modelling through the representation of cloud processes.



# 1 Introduction

Clouds in the upper troposphere play an important role in climatic processes, interacting with long wave radiation leaving the Earth's troposphere, controlling moisture entering the stratosphere and ensuring the overall stability of the Earth's atmosphere (Fueglistaler et al., 2009). Cirrus clouds are often modelled using homogeneous ice nucleation theory (Kärcher, 2002; Kärcher and Lohmann, 2002). However, in-situ studies of low temperature and subvisible cirrus clouds, which exist in the upper troposphere near to the tropical tropopause, have found evidence of fewer than expected ice crystal numbers and higher than expected supersaturations with respect to ice (Krämer et al., 2009; Jensen et al., 2005, 2013, 2017). These observations suggest that our current understanding of the ice formation mechanisms and therefore methods of modelling the formation of low temperature cirrus clouds are incorrect or incomplete (Peter et al., 2006; Krämer et al., 2009; Jensen et al., 2010). A number of different mechanisms have been proposed for the discrepancies observed between atmospheric observations of ice number densities and ice supersaturations with those predicted in model simulations, which can be described by either a suppression in the rate of ice nucleation or by a suppression in ice crystal growth (Peter et al., 2006). There are relatively few studies that have been able to characterise both physical and chemical aerosol properties in the tropical tropopause layer due to it's inaccessibility and magnitude (Lawson et al., 2008; Krämer et al., 2009; Jensen et al., 2017). However, aerosol particles in the region were found to be composed of an internal mixture of organic material and sulphates, based on the composition of ice residuals collected from subvisible cirrus clouds (Froyd et al., 2010).

Since the discovery that organic aerosol particles could exist in an amorphous or viscous state under atmospheric conditions (Zobrist et al., 2008; Virtanen et al., 2010), there has been a push to better predict the atmospheric implications of aersosol phase (Reid et al., 2018) and to characterise the global phase distributions of aerosol particles (Shiraiwa et al., 2017; Maclean et al., 2017). Modelling studies have suggested that in the cold, dry regions of the upper troposphere, aerosols were most likely to exist in a glassy state (Shiraiwa et al., 2017). Hence the ice nucleating ability or inability of viscous biogenic secondary organic aerosols could provide a crucial mechanism for the formation of cirrus clouds high in the tropical tropause (Froyd et al., 2010; Wilson et al., 2012).

The role of organic aerosol in the formation of ice has been a rapidly progressing area of research since the publication of a review of laboratory experiments (Hoose and Möhler, 2012). In particular, the role of viscous organic aerosol on ice nucleation at low temperatures has been a contested subject (Wagner et al., 2017; Reid et al., 2018). Initially, laboratory studies suggested that viscous aerosol may inhibit homogeneous ice nucleation (Zobrist et al., 2008; Murray, 2008; Murray and Bertram, 2008), however further cloud chamber studies have concluded that glassy aerosol cores could act as heterogeneous ice nuclei (Murray et al., 2010; Wang et al., 2012; Baustian et al., 2013; Berkemeier et al., 2014; Ignatius et al., 2016). $\alpha$-pinene is a biogenic precursor to secondary organic aerosol (SOA), creating particles that have the potential to exist in an amorphous state under atmospheric conditions. Experiments testing the ice nucleating ability of $\alpha$-pinene derived secondary organic aerosol have produced conflicting results, both through homogeneous nucleation (Möhler et al., 2008; Ladino et al., 2014; Wagner et al., 2017) and heterogeneous nucleation (Wagner et al., 2017; Ignatius et al., 2016). However, it is important to acknowledge that in these experiments, secondary organic aerosols were produced through various methods, including ozonolysis and OH



oxidation of $\alpha$-pinene, therefore, both chemical and physical properties of the aerosol could vary between studies (Huang et al., 2018).

A combination of laboratory studies (Lienhard et al., 2015; Price et al., 2015) and computational models (Zobrist et al., 2011; Shiraiwa et al., 2013; O'Meara et al., 2016) have used equilibration times though viscous secondary organic aerosol particles, produced from the oxidation products of $\alpha$-pinene, to extract diffusion coefficients at low temperatures through inverse modelling. Diffusion coefficients were then used in single particle simulations to interpret how equilibration timescales could interact with micro-physical cloud properties and modes of ice formation (Lienhard et al., 2015). At low temperatures, less than 195K, the viscosity of $\alpha$-pinene SOA particles is such that homogeneous ice nucleation may be inhibited due to restricted particle growth (Lienhard et al., 2015). However, at temperatures greater than 200K and low cooling rates, less than 1K s$^{-1}$, $\alpha$-pinene SOA are expected to be in equilibrium with atmospheric conditions and to freeze homogeneously. Further studies have suggested that between the temperatures of 195K and 220K, deliquescence is slowed to the extent where a glassy core forms and may act as a heterogeneous ice nucleus (Price et al., 2015).

Measurements of ice formation in chamber studies do not automatically disclose the mechanism of ice nucleation, rather, the relative humidity and temperature conditions will indicate whether ice has formed above or below the limit of homogeneous freezing (Koop et al., 2000; Wagner et al., 2017). Therefore, there is a need for modelling studies to be developed and allow comparison with the chamber studies to ensure that the underlying physical mechanisms producing ice are understood. However, only recently have computational models been developed to treat diffusion through a distribution of aerosol particles (Zaveri et al., 2018), and therefore direct comparisons to cloud chamber studies on the ice nucleating ability of viscous aerosol particles have not been attempted.

The task of this study is to develop a simple cloud parcel model, that takes into account water transport through particle bulk using an aerosol diffusion framework (Fowler et al., 2018) to better understand the effect of aerosol viscosity on ice nucleation. To appreciate the possible atmospheric implications of aerosol viscosity on ice nucleation, atmospherically relevant diffusion coefficients that depend on both temperature and water content should be used to ensure that evolving particle growth and morphology is taken into account (Lienhard et al., 2015; O'Meara et al., 2016). The overall aim of this study is to better understand the underlying physical mechanisms of ice formation in low temperature cirrus clouds and consider whether by including aerosol viscosity effects, our model is better able to simulate their formation high in the troposphere.

The article contains a short model description, followed by a methodology detailing the range of initial conditions used to test the model. To consider whether the new modelling approached is appropriate for gaining a better insight into ice formation mechanisms in amorphous aerosol particles at low temperatures, we ensure that the initial conditions used in the study are comparable to both in-situ and laboratory studies (Möhler et al., 2008; Wagner et al., 2017; Yu et al., 2017). Then the sensitivity of model simulations over the entire range of initial conditions are given in the results, followed by a few specific examples to highlight subtleties observed during extensive model testing. Finally, the implications for the way we understand cirrus clouds to form at low temperatures are discussed and brought together with conclusions.





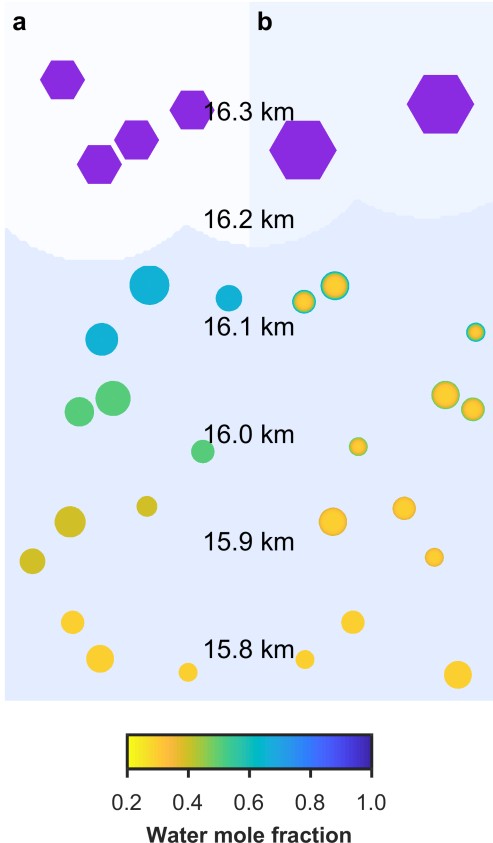

**Figure 1.** A schematic of changing particle composition in a rising cloud parcel initiated with viscous $\alpha$-pinene SOA particles, where the water mole fraction of condensed phase particles is given by the colour bar and purple hexagons represent frozen particles in a low temperature cirrus cloud. Panel **a** is the control model output, which does not take into account condensed phase diffusion and **b** is the output from the new bin micro-physics cloud model, which includes condensed phase diffusion through individual aerosol particles.

## 2   Model description

Two different models are referred to in this study; these are the new model and control model. The new model has been developed to investigate the effect of aerosol viscosity on ice nucleation and is the first to solve condensed phase diffusion of water through individual aerosol particles within a particle size-resolved cloud parcel model. The control model is also a particle size-resolved cloud parcel model, but assumes that individual aerosol particles are uniformly mixed and does not solve for condensed phase diffusion. The schematic in Figure 1a and 1b correspond to the control model and the new model including diffusion respectively and show the effect of slowed condensed phase diffusion on particle morphology and growth, hence possibly affecting the number and size of ice crystals nucleated in the simulation.



The models are initiated with a log-normal size distribution (Seinfeld and Pandis, 2006),

$$\frac{dN}{d\ln r} = \frac{N_T}{\sqrt{2\pi}\ln\sigma} \exp\left(-\frac{\ln^2\left(\frac{r}{r_m}\right)}{2\ln\sigma^2}\right), \tag{1}$$

where $N$ is the number density of aerosol particles, $r$ is particle radius, $N_T$ is the total number of aerosol particles, $r_m$ is the median radius and $\sigma$ is the geometric standard deviation of the logarithmic distribution, details of which are found in the

methods section. At the start of the simulation, the individual aerosol size bins are assumed to be in equilibrium with the surrounding gas phase. It is at this stage that the new model initiates each aerosol size bin with a diffusion framework of logarithmically spaced shells (Fowler et al., 2018).

Both models solve a system of four coupled differential equations for height, $z$, water vapour mass mixing ratio, $r_v$, pressure, $p$, and temperature, $T$, for a rising moist air parcel in hydrostatic balance. However, the new and control model differ in their

calculation of the droplet growth rate (Pruppacher and Klett, 2010),

$$\frac{dm_{w,i}}{dt} = \frac{4\pi r_i D^*(S - S_{eq,i})e_s\rho_i}{\frac{D^*L_v S_{eq,i}e_s\rho_w}{k^*T}\left(\frac{L_v M_w}{RT} - 1\right) + \frac{\rho_w RT}{M_w}}, \tag{2}$$

where $m_{i,w}$ the mass of water in each size bin $i$, $t$ is time, $r_i$ the radius of the particles in bin $i$, $S$ is the saturation ratio of water vapour, $S_{eq,i}$ is the equilibrium vapour pressure of aerosol bin $i$ with respect to liquid, $\rho_i$ is the particle density in bin $i$, $\rho_w$ is the density of water, $L_v$ the latent heat of vaporisation, $e_s$ is the saturation vapour pressure, $T$ is temperature, $R$ is the gas constant

and $M_w$ is the molar mass of water. The effects of mass accommodation are taken into account through a modified diffusivity and conductivity (Pruppacher and Klett, 2010). The two models differ in the calculation of the droplet growth because the equilibrium saturation ratio, $S_{eq,i}$, is found at the droplet's surface, which depends upon particle morphology. In the control model case, concentration is uniform throughout the particle radius and at the surface. The new model solves for condensed phase diffusion, which means concentration can vary along particle radius.

As well as the difference in calculation of the droplet growth equation, through the equilibrium saturation ratio, there are a couple of other differences between the two models. The new model assumes particle viscosity does not affect it's ability to take up water through the surface accommodation coefficient because the diffusional limit of condensation is removed when water is able to mix through the outer aerosol layers (Rothfuss et al., 2018). The second difference is that the new model calculates the nucleation rate in the individual concentric aerosol shells, which means freezing can occur in layers of high

water mole fraction on the surface of an amorphous aerosol core. The rate of homogeneous ice nucleation is calculated using temperature and the water activity of the mixture in both models (Koop et al., 2000).

For this study, Fick's second law of diffusion (Fick, 1855),

$$\frac{\partial c}{\partial t} = \frac{1}{r^2}\frac{\partial}{\partial r}\left(Dr^2\frac{\partial c}{\partial r}\right), \tag{3}$$

where $c$ is concentration, $D$ the condensed phase diffusion coefficient, $r$ the particle radius, is solved numerically for each

individual aerosol size bin using the backward Euler method (Fowler et al., 2018). Initially the Euler backward method was chosen because it conserves mass and is stable over the longer time periods associated with cloud development. However,



**Table 1.** Parameter space of atmospheric conditions used to test and compare results from the new and control models.

| Variable | Range | Reference |
|---|---|---|
| Temperature | 185-225 K | Reverdy et al. (2012) |
| Pressure | 150-70 hPa | Fueglistaler et al. (2009) |
| RH | 0.3-0.8 | Jensen et al. (2017) |
| Height | 14-18 km | Fueglistaler et al. (2009) |
| Up-draft velocity | 0.1-1 m.s$^{-1}$ | Kärcher and Ström (2003) |

during model testing it became apparent that diffusion would need to be solved simultaneously with droplet growth, and sufficiently small time-steps would be needed to ensure model stability. Therefore, it is expected that alternative diffusion frameworks incorporated into the bin parcel model would give similar solutions as little variability was found in a study comparing the sensitivity of diffusion timescales to the model framework (O'Meara et al., 2016).

## 3   Method

Throughout our study, simulations from the new model are compared with a control calculation, which does not include an aerosol diffusion framework and assumes individual particle bins have a constant concentration along the particle radius. The comparison between the new and control models allows us to assess the impact on the current understanding of ice nucleation processes and the specific conditions where aerosol viscosity has the greatest influence on cloud micro-physics. The method section contains details of initial conditions and the metric defined as the fractional difference in ice nucleation used to compare the output from the new and control models.

### 3.1   Initial conditions

Initial conditions for model simulations were selected based on atmospheric conditions in the tropical tropopause layer, where secondary organic aerosols are likely to exist in glassy and amorphous states (Zobrist et al., 2008; Virtanen et al., 2010; Shiraiwa et al., 2017). Table 1 shows the full parameter space of atmospheric conditions tested with the cloud parcel models.

Figure 2 shows three aerosol size distributions used to initialise model simulations. Size distribution 1 corresponds to organic aerosol data collected near the tropical tropopause (Yu et al., 2017) and size distributions 2 and 3 correspond to two laboratory studies investigating the ice nucleating ability of $\alpha$-pinene SOA (Möhler et al., 2008; Wagner et al., 2017). Although size distributions 2 and 3 are from laboratory studies, extremely high number concentrations of organic aerosol have been observed in the upper troposphere near to regions of deep convective outflow in the tropics (Andreae et al., 2018). The molecular weight (136.2 g.mol$^{-1}$) and density (1260 kg.m$^{-3}$) of $\alpha$-pinene SOA were kept constant throughout the study.





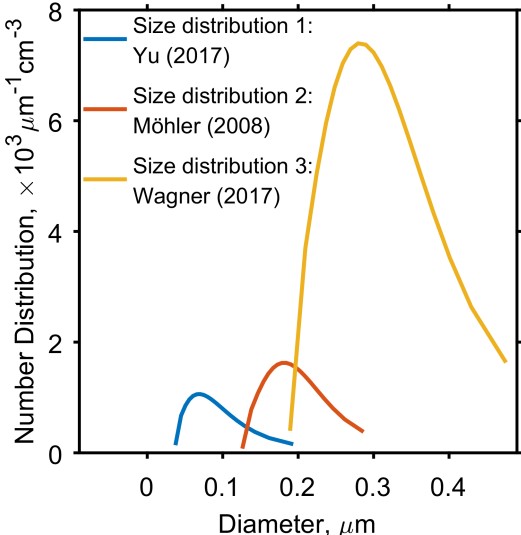

**Figure 2.** Three aerosol particle size distributions used to initiate the cloud parcel models, which have been taken from published studies. Yu et al. (2017), corresponds to secondary organic aerosol data collected near the tropopause in Kumming, China and both Möhler et al. (2008) and Wagner et al. (2017) correspond to aerosol size distributions used in cloud chamber experiments. Table A1 found in the appendix gives specific parameter values for the log-normal size distributions.

A number of studies have aimed to quantify the effect of aerosol viscosity on equilibration timescales through $\alpha$-pinene SOA by estimating coefficients of water diffusion (Renbaum-Wolff et al., 2013; Price et al., 2015; Lienhard et al., 2015). The water diffusion coefficients for $\alpha$-pinene SOA from Lienhard et al. (2015) were specifically used in this study because the parameterisation is valid down to the low temperatures observed high in the troposphere and covers the large temperature range appropriate for our experiment as indicated by the initial conditions in Table 1. Figure 3 shows the diffusion coefficient as a function of temperature and water activity; highlighting how temperature conditions for homogeneous freezing, shown by the grey shaded region, interact with the published $\alpha$-pinene SOA diffusion coefficients. Figure 3 indicates that at low temperatures, below 200K, slow diffusion could influence homogeneous ice formation. It is important to acknowledge that published $\alpha$-pinene SOA diffusion coefficients vary by orders of magnitude (Price et al., 2015) and that aerosols produced in laboratory studies may differ in composition to those produced in the atmosphere (Huang et al., 2018). The model simulations are initiated just below cloud base so we envisage that there will be no further co-condensation of volatile vapours with decreasing temperature as the cloud parcel begins to rise.





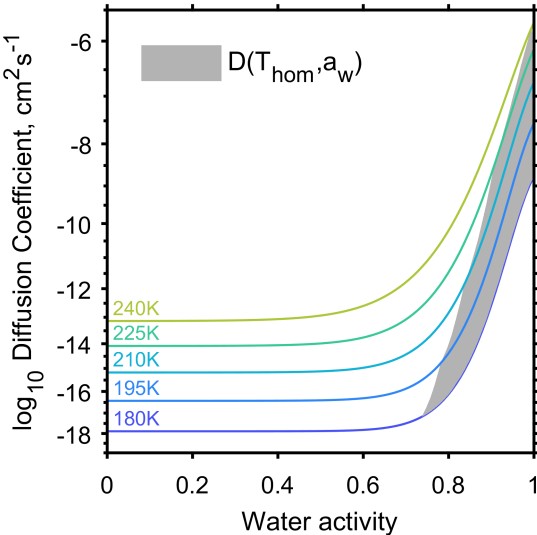

**Figure 3.** Water diffusion coefficient through $\alpha$-pinene secondary organic aerosol as a function of temperature and water activity adapted from *Viscous organic aerosol particles in the upper troposphere: diffusivity-controlled water uptake and ice nucleation?* by Lienhard et al. (2015). The grey region shows where the conditions of homogeneous freezing and diffusion coefficient interact.

## 3.2 Metric for model testing

To compare the number of ice crystals formed by the new and control model we chose to define the fractional difference between the number of ice produced by the two simulations to be,

$$\text{fractional difference in } N_{\text{ice}} = \frac{N_{\text{ice}}^{\text{new}} - N_{\text{ice}}^{\text{control}}}{N_{\text{ice}}^{\text{new}} + N_{\text{ice}}^{\text{control}}}, \tag{4}$$

5  where, $N_{\text{ice}}^{\text{new}}$ is the number density of ice produced by our new model and $N_{\text{ice}}^{\text{control}}$ is the number density of ice produced by the control model. The fractional difference in $N_{\text{ice}}$ was chosen as the metric to compare ice nucleation rates, as the values are limited between -1 for no ice produced in the new model and 1 no ice produced by the control model.

## 4 Results

### 4.1 Model sensitivities

10  The results of this study begin with a sensitivity analysis, comparing the outputs from both the new and control models to investigate how specific input variables effect aerosol viscosity and ice formation. The parameter space which the models are tested over are introduced in Table 1. Both rates of diffusion and ice nucleation depend upon temperature and water activity or concentration, therefore much of the discussion in this section concentrates on factors that directly or indirectly the temperature and the rate of water uptake by the individual aerosol size bins.



Figure 4 shows three different parcel trajectories: typical cirrus ($T \sim 225$K), colder cirrus ($T \sim 210$K), and very cold cirrus ($T \sim 200$K). As the cloud parcels rise in simulations, the temperature decreases and ice saturation ratio increases until conditions for homogeneous ice nucleation are met. As aerosol particles freeze and ice crystals grow, the parcel of air dehydrates, shown by the rapid decrease in ice saturation ratio. The fractional difference in the number of ice crystals frozen between the

new and control model simulations are given as a function of the initial temperature and ice saturation conditions. The blue colours represent a suppression in the number of ice crystals formed by our new model in comparison to the control model, and red colours an enhancement in the number of ice crystals formed by our new model in comparison to the control model. For each cloud parcel simulation in Figure 4, a representative aerosol particle concentration profile shows that as initial temperature decreases, the equilibration timescale increases.

In model simulations initiated under very cold cirrus conditions, Figure 4 shows a very thin layer of water on the outside of the individual particles. The immobility of water molecules through the aerosol particle is due to the high viscosity or small diffusion coefficients of water through $\alpha$-pinene SOA at temperatures below 200K. The small volume of water on the aerosol particles in the cold temperature simulations results in a suppression of ice formation rate in comparison to the control model simulations. Cold cloud chamber studies investigating the ice nucleating ability of $\alpha$-pinene SOA have not been taken down

to the temperatures used in these model simulations. However, inhibited homogeneous ice nucleation at high cooling rates has been observed in citric acid particles (Murray, 2008), which are viscous at higher temperatures than $\alpha$-pinene SOA (Price et al., 2014). In the cold cirrus case, aerosol particles are less viscous and water molecules more mobile, which means there is a larger layer of water on the surface of the aerosol particles. The larger volume of high water activity in the outer layer of aerosol particles results in an enhanced rate of ice formation in comparison to the control model. Under the temperature

conditions of a typical cirrus cloud, we predict that $\alpha$-pinene SOA particles are almost in equilibrium with their surroundings by the time conditions for homogeneous ice nucleation are reached and only a small aerosol core still visible in the larger particles of the size distribution. Hence, under typical cirrus conditions the models do not predict any difference in the number of aerosol particles frozen whether viscosity or accounted for or not.

   Aerosol particles in all three parcel trajectories from Figure 4 exhibit a boundary between the outer layer of water and the

highly viscous $\alpha$-pinene SOA core, represented by the contrast in water mole fraction. These steep gradients in concentration between the core and outer layer of water have been observed using Mie resonances in levitated highly viscous aerosol particles (Bastelberger et al., 2018) and reproduced in model simulations (O'Meara et al., 2016). Diffusion fronts are formed as a result of the plasticising effect of water, where a small increase in the volume of water greatly reduces the viscosity of a solution. The evolution of the diffusion front through the aerosol particle is limited by the aerosol viscosity ahead of the diffusion front.

The diffusion front is recreated by the model using a sigmoidal relationship between water activity and diffusion coefficient for $\alpha$-pinene SOA (Lienhard et al., 2015).

   Current models simulate the changing composition of a single viscous aerosol particle under changing atmospheric conditions (Lienhard et al., 2015), however, our new model allows the cumulative effect of particle viscosity on ice formation within a cloud parcel to be studied. Figure 5 extends the investigation of the effect of particle viscosity on ice nucleation, by

taking into account both up-draft velocity and aerosol size distribution. The fractional difference in the number of ice crystals





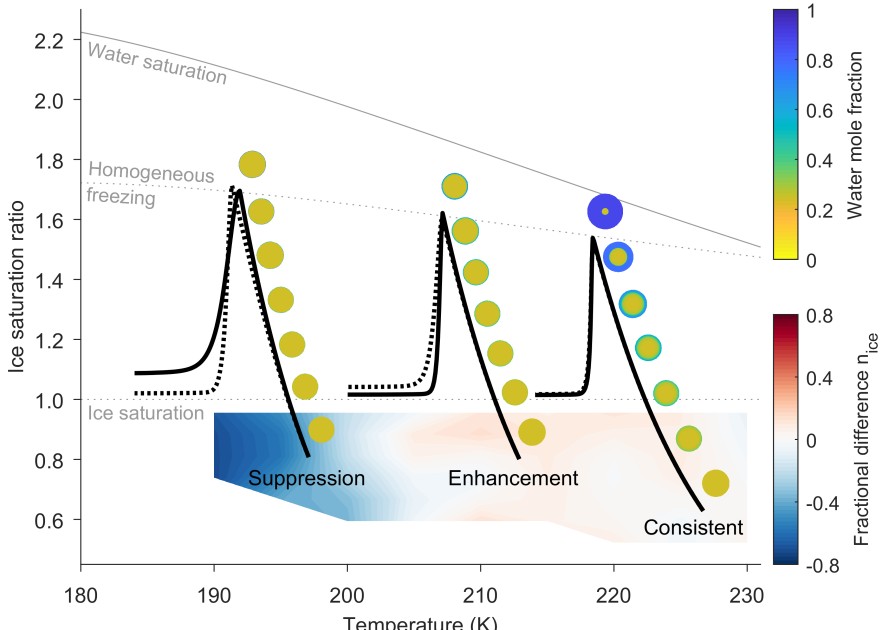

**Figure 4.** A schematic showing the effect of initial temperature on particle composition, ice formation and ice saturation ratio, highlighting the effects of temperature dependant diffusion on ice nucleation for three specific cloud parcel trajectories. The solid black lines show a cloud parcel trajectory for our new model and the broken black line for the control simulation. The changing water mole fraction for a single particle is shown next to each of the parcel trajectories. The fractional difference in number of frozen particles is shown at the initial conditions of ice saturation ratio and temperature, where the blue areas show that ice formation is suppressed in the new model simulations in comparison to the control model and red regions show an enhancement in the number of frozen particles in new model simulations. At the lowest temperatures, below 200K water molecules are immobile and particle growth is restricted, leading to a suppression in ice formation. As temperature increases, water molecules become more mobile and diffuse through the viscous SOA particle, so that between 200K and 220K there is an increase in the number of particles frozen as a result of the thicker layer of water on the aerosol surface. At higher temperatures, above 220K an aerosol core is still visible, however the core shrinks in relation to the outer liquid water layer until the aerosol particle is almost totally mixed as the parcel reaches the conditions for homogeneous freezing, which results in a consistent number of particles freezing in both the new and control model. Simulations from both the new and control model have been initiated with a size distribution 3, given in Figure 2, under a constant up-draft velocity of $0.6 \mathrm{ms}^{-1}$.

nucleated between the bin diffusion model and a control model are given as a function of initial temperature and ice saturation ratio. As with Figure 4, blue colours represent a suppression in the number of ice crystals by our new model compared to control model runs, and red an enhancement in the number of ice crystals by the new model compared to control model runs. To reproduce the conditions of low temperature and subvisible cirrus clouds, the simulations are initiated with temperatures,

5  pressures, relative humidities and up-draft velocities typically found in the tropical tropopause layer from Table 1.



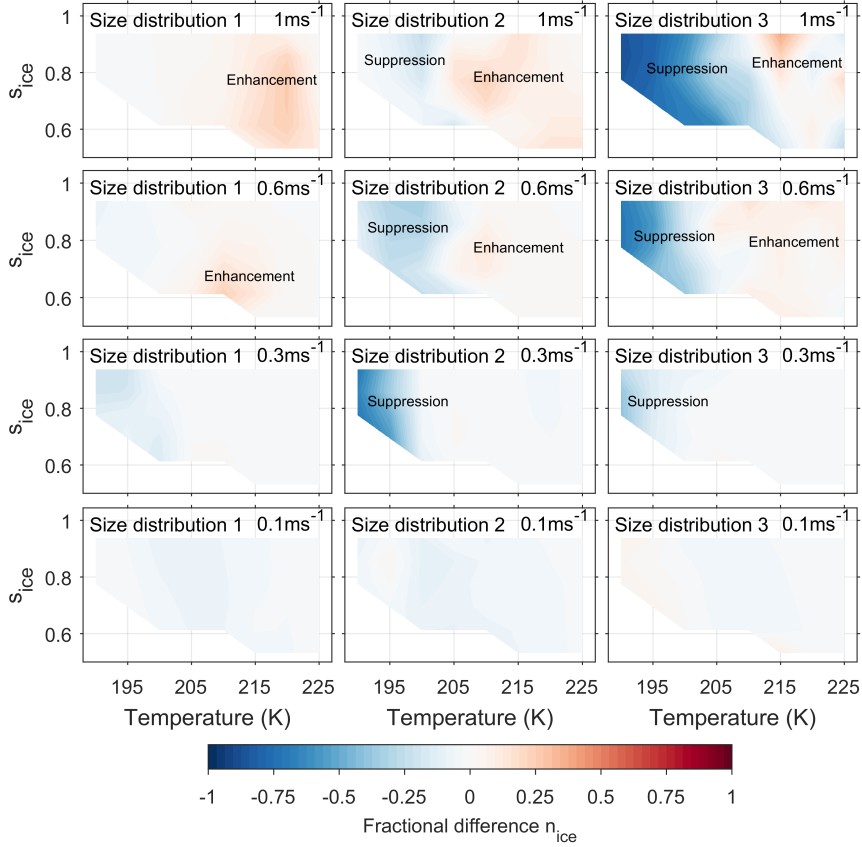

**Figure 5.** Shows the effect of initial conditions on the fractional difference in number of frozen particles between the new and control model simulations. The blue areas show that ice formation is suppressed in the new model simulations in comparison to the control model and red regions show an enhancement in the number of frozen particles in new model simulations. The models have been initiated with three different size distributions, which are given in Figure 2, and with constant up-draft velocities ranging from $0.1\text{ms}^{-1}$ to $1\text{ms}^{-1}$. Figures showing the number density of ice crystals formed in the new and the control model simulations are available in the appendices.

As up-draft velocity increases in Figure 5 there is a greater fractional difference in the number frozen aerosol particles between the new model and control model simulations. At higher up-draft speeds there is less time for the diffusion front to move through the aerosol to its centre, therefore the viscous $\alpha$-pinene SOA particles do not have time to equilibrate with parcel conditions. Hence at up-draft speeds of $0.6\text{ms}^{-1}$ or greater, both the low temperature suppression of ice nucleation and the higher temperature enhancement of ice nucleation are amplified. In the case of size distribution 3, a fractional difference in the number of ice crystals of up to 0.8 is found, which corresponds to a suppression of over $50\text{cm}^{-3}$ of ice between the new and control model. The enhancement of ice nucleation by $\alpha$-pinene SOA at temperatures greater than 200K or the colder cirrus conditions, only occurs at up-draft velocities of $0.6\text{ms}^{-1}$ or greater. The enhancement is a result of a layer of water forming on the viscous aerosol core. For the number of ice crystals formed by the new model to be greater than the number produced by





the control model conditions need to be optimal, allowing for enough liquid water to condense for homogeneous ice nucleation to occur in the outer liquid water layer. At lower up-draft velocities, $0.3\mathrm{ms}^{-1}$ and below, the diffusion front moves through the particle before the conditions for homogeneous ice nucleation are reached, therefore an enhancement in the number of frozen particles does not occur. At the lowest up-draft velocities of $0.1\mathrm{ms}^{-1}$, viscosity has much less of an effect on ice nucleation

given that the fractional difference in frozen particles is less than 0.1. At low up-draft velocities droplet growth is predicted to be similar in both the new and control models as viscous particles have plenty of time to equilibrate with the parcel conditions, which results in a very similar number of ice crystals forming.

Figure 5 shows an enhanced suppression of ice nucleation in the simulations initiated with the size distribution 3 compared size distributions 1 and 2. We suggest that the amplified suppression of ice by the new model simulations in comparison to the

control for size distribution 3 is because condensed phase water is distributed between a greater number of particles in the high aerosol number concentrations as indicated by Figure 2. Therefore, under the very cold cirrus conditions, or the most viscous cases, less liquid water condenses on the outer layer of individual aerosol particles and fewer aerosol particles freeze.

Figure 5 shows that the smallest fractional difference in the number of frozen aerosol particles between the new and control models were in simulations initiated with size distribution 1. There are a couple of reasons why aerosol viscosity has a limited

effect on ice nucleation from the in-situ case of size distribution 1 (Yu et al., 2017). The first is the smaller number concentration of aerosol particles, shown in Figure 2, which means liquid water is distributed between fewer aerosol particles resulting in a higher water activity in the outer aerosol shells. The high water activity in the outer shells causes particles to equilibrate before the conditions for homogeneous ice nucleation are reached, due to the plasticising effect of water on viscous aerosols (O'Meara et al., 2016). Secondly, the median diameter of size distribution 1 is the smallest used in this study. A particle with a smaller

diameter is expected to equilibrate with the gas phase in a shorter amount of time in comparison to a larger particle, since diffusion timescales are proportional to the square of the diameter over the diffusion coefficient (Price et al., 2015). Therefore, simulations initiated with size distribution 1 are likely to have equilibrated before reaching the conditions of homogeneous ice nucleation. Our findings suggest that viscous organic aerosol, such as $\alpha$-pinene SOA, would have a limited effect on the rate of homogeneous ice nucleation in regions of low number concentrations, even at the coldest temperatures, below 200K, found in

the tropical tropopause layer. Previous studies using single particle simulations have also concluded that aerosol viscosity may have a limited effect on homogeneous ice nucleation under certain atmospheric conditions (Lienhard et al., 2015). However, the investigation into the effect of viscous aerosol on ice nucleation in low temperature cirrus should not end there as the new model could be used to gain further insights into how cloud processing could make alternative ice nucleation mechanisms possible under atmospheric conditions.

## 4.2   Processing of viscous aerosol particles

Figure 6 shows how ice saturation ratio and number of ice crystals formed depends upon cloud parcel temperature for four model simulations, all initiated on the same parcel trajectory but at different initial ice saturation ratios. Only one control model run, shown by the black line, has been plotted on the graph, since both the ice saturation ratio and number of ice crystals formed are consistent for the different initial ice saturation ratios. The upper panel shows that parcels initiated in equilibrium





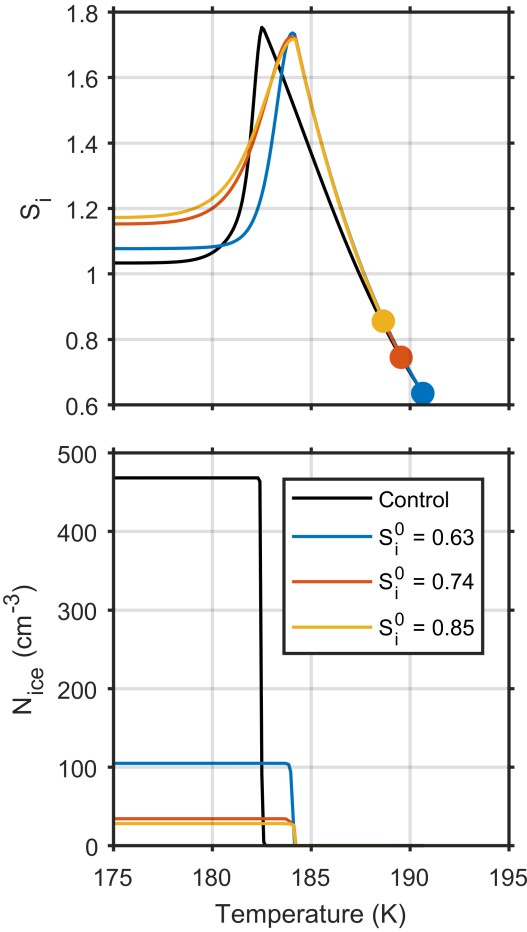

**Figure 6.** The effect of different initial ice supersaturations, $S_i$, on model simulations.

with atmospheric conditions, closer to cloud base result in a higher steady state ice saturation ratio after homogeneous freezing takes place, which is a result of fewer aerosol particles freezing, as shown in the lower panel. Figure 6 contains simple plots, but they show that as the timescales of diffusion become a limiting factor in cloud droplet growth, the history of individual particles needs to be considered in relation to parcel vapour content and ice nucleation.

5    Pre-activation of ice nuclei has been discussed in relation to the modes of ice nucleation in amorphous secondary organic aerosols (Wagner et al., 2012) and in particular $\alpha$-pinene SOA (Wagner et al., 2017; Ladino et al., 2014). Studies have suggested that during a drying cycle, when water is removed from the outer layer of an aerosol particle that an amorphous shell could form trapping water in aerosol core to later freeze heterogeneously through inside-out contact nucleation (Durant and Shaw, 2005). An alternative suggestion is that as water is removed, ice residuals could be left inside the core of the particle or within

10   a glassy porous structure leading to a nucleation event (Wagner et al., 2017). We did not intend to investigate pre-activation, however on closer inspection of model runs initiated with low up-draft speeds ($0.1 \mathrm{ms}^{-1}$), such as Figure 7, we noticed that



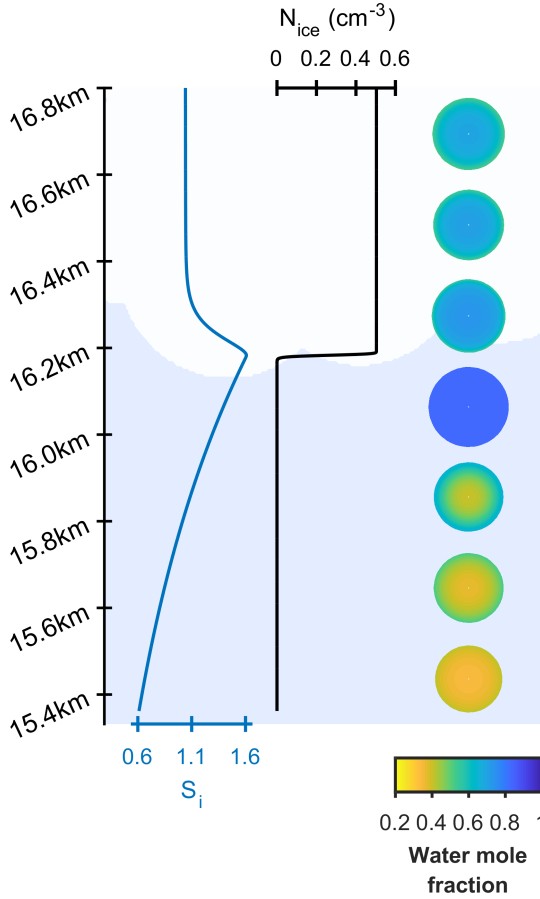

**Figure 7.** A schematic showing the changing aerosol particle morphology after homogeneous ice nucleation.

after a freezing event and the drop in the cloud ice saturation ratio, water was removed from the outer layers of the aerosol particle, leaving high water activity in the aerosol core. In noticing the formation of an organic shell we wonder if under further model development, the effect of aerosol viscosity on ice nucleation during multiple cooling cycles could be investigated.

## 5 Atmospheric implications

5  The current understanding of ambient aerosol particle viscosity and the implications this has for atmospheric processes is underpinned by laboratory studies (Reid et al., 2018). These laboratory studies have enabled the researchers to probe single aerosol particles both as samples deposited of substrates and as levitated particles over a wide range of temperature and viscosity (Krieger et al., 2012). However, few studies have investigated the effect of viscosity on a population of aerosol particles (Yli-Juuti et al., 2017; Ye et al., 2016; Zaveri et al., 2018). Using the new cloud parcel model that solves for condensed





phase diffusion through individual aerosol particles enables a different insight to the atmospheric implications of viscous aerosol on ice nucleation processes.

The suppression of homogeneous ice nucleation at low temperatures by ultra viscous aerosol has been suggested as a reason for the high in cloud supersaturations observed in low temperature cirrus clouds or subvisible cirrus (Murray, 2008; Peter et al., 2006; Krämer et al., 2009; Froyd et al., 2010). Figure 4 shows both the parcel trajectories for our new model and control simulations, which are given by the solid and broken black lines respectively. In the highly viscous regime, where model simulations are initiated below 200K, the suppression in the number of ice particles by the new model causes the parcel of air to dehydrate at a slower rate in comparison to the control model, which can be seen in the gradient of the parcel trajectories after ice nucleation. There is a clear difference between the steady state relative humidities with respect to ice, where the control model predicts a value close to ice saturation and the new model predicts an ice saturation ratio of 1.1.

In this study we consider the effect of aerosol particle viscosity on the homogeneous nucleation of ice; however, it is important to acknowledge that there is evidence that viscous aerosols can nucleate ice heterogeneously. Laboratory experiments have shown that a variety of viscous substances nucleated ice well below the homogeneous limit in a cloud chamber study (Murray et al., 2010; Wilson et al., 2012), and a more recent study has found evidence that $\alpha$-pinene SOA also acts as a heterogeneous ice nucleus (Ignatius et al., 2016).

The modes of heterogeneous ice nucleation are less well understood and modelled in comparison to homogeneous ice nucleation. Despite this, under low temperature cirrus conditions, where highly viscous organic aerosols can exist in a glassy phase, there is evidence that these particles could nucleate ice in the deposition mode of freezing (Wang et al., 2012). As temperatures increase, water molecules become more mobile forming a thicker outer shell with a high water activity, in this regime it has been suggested that ice nucleation could occur through the immersion mode on the viscous organic aerosol core (Murray et al., 2010). Using diffusion coefficients, our model replicates the cold and warm regimes observed in laboratory experiments through totally inhibited growth and the formation of a condensed liquid phase outer shell. Other studies have investigated the regimes of ice nucleation under cold cirrus conditions using metrics such as glass transition temperature (Wang et al., 2012), the limit between an amorphous substance being in a glassy state and a viscous state.

To better understand how our approach differs with standard methods of modelling cirrus cloud development in the upper troposphere, we compare cloud parcel simulations with a parameterisation for homogeneous ice formation used in large scale general circulation models (Kärcher and Lohmann, 2002). Figure 8 shows how ice crystal number concentrations depend upon temperature and up-draft. Model simulations have been initiated just below ice saturation, with an aerosol number density of $1000 \text{cm}^{-3}$, at temperatures of 230K, 210K and 190K, and a range of constant up-draft speeds from 0.01 to $1 \text{ms}^{-1}$. At low up-draft speeds, where aerosol particles are able to equilibrate with the surrounding gas, the number density of ice crystals formed in both the models and predicted by the parameterisation agree. However, at higher up-draft speeds, greater than $0.5 \text{ms}^{-1}$, we notice that the number of ice crystals formed by the new model, initiated at 190K, deviate from both the control and the general circulation model paramterisation by an order of magnitude. The trend seen in Figure 8 is similar to observed ice crystal number concentrations, where the number of ice crystals measured ranged from 0.005 to $60 \text{cm}^{-3}$ (Krämer et al., 2009). By





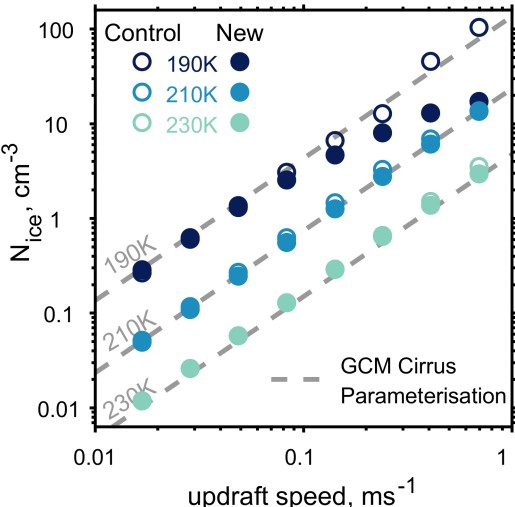

**Figure 8.** Showing the effect of up-draft speed and temperature on the formation of ice for both the new and control cloud parcel models is shown by the filled and outlined circles respectively. The grey dashed line represents a parameterisation for ice number density in cirrus clouds used in a large scale general circulation model (Kärcher and Lohmann, 2002).

taking into account particle viscosity, through condensed phased diffusion we better represent atmospheric observations of ice crystal numbers in cirrus than are currently used in large scale general circulation models.

## 6 Conclusions

The model analysis presented in this paper suggests that there are two competing effects as a result of the immobility of

5 water molecules through viscous $\alpha$-pinene secondary organic aerosol particles. The first is the inhibition of homogeneous ice nucleation at the lowest temperatures, below 200K, due to restricted particle growth and low water volume. The second effect occurs at warmer temperatures, between 200K and 220K, where water molecules are more mobile and a layer of water condenses on the outside of the particle, leading to an increase in the homogeneous nucleation rate. The new cloud parcel model that solves for condensed phase diffusion through individual aerosol particles provides a platform to further investigate

the effect of condensed phase diffusion on cloud micro-physical processes and address some of the problems associated with modelling cirrus clouds in larger scale general circulation models. Initial research, addressed in this study, has focused on how viscous secondary organic aerosol particles, such as $\alpha$-pinene SOA, effects ice nucleation in low temperature cirrus clouds. However, recent work has highlighted that anthropogenic sources also produce highly viscous aerosols that exist at much higher temperatures and relative humidities (Reid et al., 2018), which could have implications under the conditions of mixed phase

cloud formation (Charnawskas et al., 2017). Then there are more complex multi-component systems to consider, containing a mixture of SOA and inorganic salts have been observed to form liquid-liquid phase separations, which can effect cloud particle



growth and activation at higher temperatures and relative humidities (Renbaum-Wolff et al., 2016). There is scope to further develop our new model to investigate how the solubility of aerosol particles within multi-component systems could affect cloud micro-physical properties using a Maxwell-Stefan diffusion framework (Fowler et al., 2018).



**Table A1.** Parameters used to plot the aerosol size distributions in Figure 2.

| Variable | Yu et al. (2017) | Möhler et al. (2008) | Wagner et al. (2017) |
|---|---|---|---|
| $N_t$ | 100 cm$^{-3}$ | 188 cm$^{-3}$ | 1500 cm$^{-3}$ |
| $r_m$ | 85 nm | 190 nm | 300 nm |
| $\sigma$ | 1.65 | 1.32 | 1.29 |
| $\ln(\sigma)$ | 0.5 | 0.28 | 0.25 |

## Appendix A:  Parameters for log-normal size distributions

Parameter values in Table A1 for $N_t$, the total number density of aerosol particles, $r_m$, the median radius and $\sigma$, the geometric standard deviation of a log-normal distribution, correspond to the aerosol size distributions used throughout the study and plotted in Figure 2.

## 5   Appendix B:  Extended results

Figures B1 and B2 have been included in the appendix to quantify the number of ice crystals nucleated by both the new model and control model respectively. The colour scale shows the frozen fraction of total aerosol particles in the simulations and contour lines give the number of particles nucleated per cubic centimetre. The fractional difference in $N_{\mathrm{ice}}$, from Equation 4 and represented by the colour scale in Figure 5, corresponds to the difference between Figures B1 and B2 divided by their sum.
10  By comparing the contour lines in the simulations run with size distribution 3 and up-draft speeds of 0.6ms$^{-1}$ and above we are able quantify the difference that viscosity could have on the number of ice crystals formed.




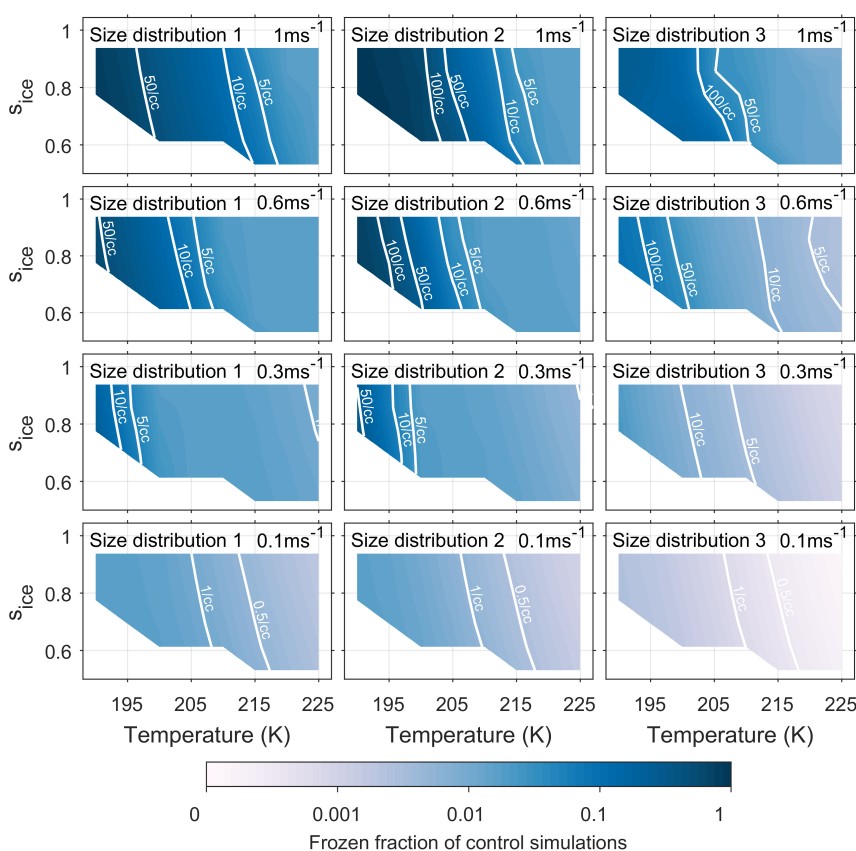

**Figure B1.** Quantifying the number of aerosol particles frozen in control model simulations. The colour scale represents the frozen fraction of total aerosol particles and the labelled contour lines give the number of frozen aerosol particles per cubic centimetre. The model run have been initiated with three different size distributions, which are given in Figure 2, and with constant up-draft velocities ranging from $0.1\,\mathrm{ms}^{-1}$ to $1\,\mathrm{ms}^{-1}$.





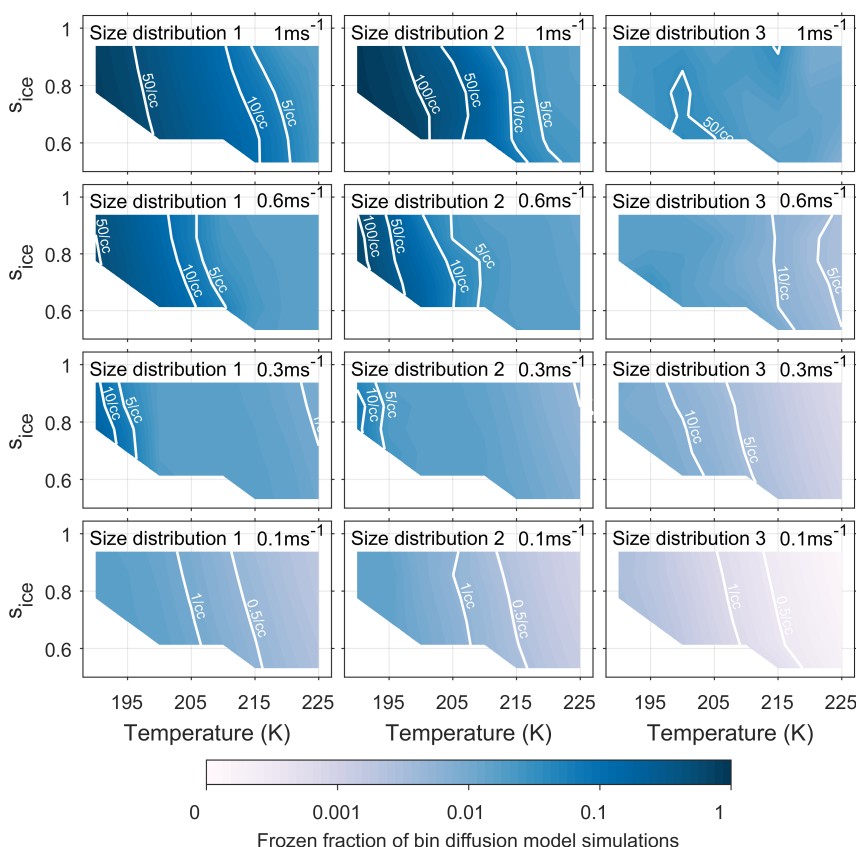

**Figure B2.** Quantifying the number of aerosol particles frozen in bin diffusion model simulations. The colour scale represents the frozen fraction of total aerosol particles and the labelled contour lines give the number of frozen aerosol particles per cubic centimetre. The model run have been initiated with three different size distributions, which are given in Figure 2, and with constant up-draft velocities ranging from $0.1 \text{ms}^{-1}$ to $1 \text{ms}^{-1}$.





*Author contributions.* PJC and KF conceived and designed the research and developed the model code. KF led the analysis and figure preparation and wrote the manuscript. KF, PJC and DT contributed to scientific discussions and to the preparation of the manuscript.

*Competing interests.* The authors declare no competing interests.

*Acknowledgements.* This publication contains work conducted during the PhD studentship of Kathryn Fowler, supported by the Natural Environment Research Council (NERC) EAO Doctoral Training Partnership and is fully-funded under the grant reference number NE/L002469/1. PJC acknowledges support from the European Union's Seventh Framework Programme (FP7/2007-2013), under grant agreement number 603445.



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
