# Peer review of "Modelling the effect of condensed-phase diffusion on the homogeneous nucleation of ice in supercooled water"

_Atmospheric Chemistry and Physics, 2019_

## Referee Comment (RC1) · Anonymous Referee #1 · 24 Jul 2019

Review of

**Modelling the effect of condensed-phase diffusion on the homogeneous nucleation of ice in supercooled water**

by Fowler et al.

**General:**
This study simulates  homogeneous ice nucleation by taking into account condensed phase diffusion through  individual ultra viscous aerosol particles by means of a new cloud parcel model with bin microphysics. The major findings of the study are that  homogeneous ice nucleation is inhibited below 200K due to restricted particle growth and low water volume, while  at higher temperatures between 200K and 220K the number of frozen aerosol particles increases because the  water molecules are slightly more mobile  and a layer of water condenses on the outside of the particle. The topic of the study is  interesting and timely, because the representation of especially ice clouds in climate models needs to be improved in order to reduce major uncertainties in the prediction of the future climate.

The paper is well organized and fluently written. I like to mention in particular that  the Figures are not only beautiful, but also prepared in a way that makes complex relationships easily accessible for the reader. The  methods applied seems sound to me, though I do not feel that I can evaluate the model framework because this is on the edge of my expertise.

My rating of the paper is, however,  'major revisions' for the reason I will explain in the following: unfortunately, the atmopheric conditions framing the paper do not match the presented results. The authors claim to help explain observations of very few ice crystals and high supersaturations in the very cold (< 205 K) tropical tropopause layer (TTL).
Observed conditions in TTL cirrus clouds are:  low ice particle concentrations around 0.005 – 0.1 cm-3, vertical velocities most frequently around a few cm/s  (or even slower), infrequent waves  up to ~2 m/s (the low/high ice  concentrations are found in the slow/fast updrafts).
The conditions where suppression of homogeneous ice nucleation is found in the paper (Figure 8) are: ice particle concentrations  > ~3 cm-3 at vertical velocities 0.1 – 1 m/s.  These vertical velocities  and thus ice concentrations represent the atmospheric range of gravity waves, e.g. behind mountains, or convection (see for  example Kärcher and Lohmann, 2002), but can not be extrapolated to TTL conditions. The results shown in Figure 8 clearly indicate that for TTL conditions the new and control simulations do not greatly differ, that means that no further understanding of the TTL ice concentrations can be obtained from this study.

Nevertheless, to better understand the processes of homogeneous ice formation is important also  in strong updrafts, simply for a correct modelling of high ice concentrations occuring at very low temperatures in many geographical regions (see e.g. Sourdeval et al., 2018 and Gryspeerdt et al., 2018, both ACP) or also  for example from the point of view of cirrus seeding. Thus, I would encourage the authors to interpret their interesting findings in relation to the corresponding atmospheric environments.

**Specific comments:**

**Title:**  From the title I did not have a good idea of the content of the paper. I would suggest to replace ‚supercooled water' by ‚ultra viscous particles', because the term ‚supercooled water'  to me implies homogeneous freezing  of liquid cloud drops at -38C.

**p 2, l 7-9:**   *'These observations suggest that our current understanding of the ice formation mechanisms and therefore methods of modelling the formation of low temperature cirrus clouds are incorrect or incomplete (Peter et al., 2006; Krämer et al., 2009 ; Jensen et al., 2010).'*
Many more publications treating the topic appeared later – in case you mention the TTL in the next version of the manuscript, the most impostant  newer studies should also be cited.

**p 6, l 20:**  *'… extremely high number concentrations of organic aerosol have been observed in the upper troposphere near to regions of deep convective outflow in the tropics (Andreae et al., 2018).'*
The observations of Andreae et al. (2018) are well below the TTL, so is this statement really relevant for the study ?

**P 10, l 5:** *'… up-draft velocities typically found in the tropical tropopause layer from Table 1.'*
The  updraft range used for the study is not typical for the TTL, in particular the large scale updraft is lower, see comments to Table 1.

**Figures 4 and 5:**
The simulations shown in Figure 4 are (as can be seen from Figure 5) those with almost the largest effect on ice nucleation (Sze distribution 3, 0.6 m/s). The interpretation of the results regarding TTL cirrus are based  mainly on this scenario. However, these conditions are untypical for the  TTL – especially the updrafts are much slower (see comment on Table 1).  For the TTL, the scenario shown  at the bottom of the left column of Figure 5 would be most appropriate, though even here the updraft is rather high (only a few cm/s is typical in the TTL).

**Figure 8:**
This Figure shows that the suppression of homogeneous ice nucleation occurs for ice  particle concentrations  larger than ~3 cm-3. The low concentrations that are under discussion, however, are in the range of 0.005 – 0.1 cm-3 (Krämer et al, 2009; Jensen et al., 2013; Spichtinger and Krämer, 2013), where no difference between the control and the new cloud parcel model is visible, or this range is not covered by the simulations.

**Table 1:**

**Table 1.** Parameter space of atmospheric conditions used to test and compare results from the new and control models.

| Variable | Range | Reference |
| --- | --- | --- |
| Temperature | 185-225 K | Reverdy et al. (2012) |
| Pressure | 150-70 hPa | Fueglistaler et al. (2009) |
| RH | 0.3-0.8 | Jensen et al. (2017) |
| Height | 14-18 km | Fueglistaler et al. (2009) |
| Up-draft velocity | 0.1-1 m.s$^{-1}$ | Kärcher and Ström (2003) |

The parameter spaces for TTL cirrus clouds specified in this table are partly not correct:
Temperature: 185 – 205 K
Updraft:        0.01 – ~2 m/s, or even slower in the TTL
                 Spichtinger and Krämer (2013), ACP, also
                 Jensen et al. (2012), JGR

---

## Referee Comment (RC2) · Anonymous Referee #2 · 30 Jul 2019

The authors extend their previous development of a single-particle aerosol model, which they used to calculate aerosol-water vapor equilibration times for individual organic aerosol particles exposed to a step change in relative humidity. Here they use the new model to calculate the response of an aerosol size distribution that is initially at equilibrium with water vapor and then ascends until homogeneous freezing leads to an approximately steady state ice crystal number concentration. They consider a range of aerosol size distribution parameters, ascent rates, and initial parcel thermodynamic states. The methodology appears sound, although I am not an expert in all details, except that some additional information is required to make the results reproducible. The conclusions appear generally reasonable, but should be placed more quantitatively into

the context of the deficiencies in current models that motivated this work. There are grammatical errors that can be readily handled by a copy editor; I only note one below where the meaning was not clear to me. To the limits of my expertise, I would rate a revised manuscript as suitable for publication in ACP if it addresses the following questions.

General comments

1. Reproducibility

1a. Is the ascent adiabatic with respect to the model parcel? If so, parcel temperature must account for the latent heat of condensation. Please explain how environmental temperature and water vapor mixing ratio are calculated during parcel ascent.

1b. Somewhere before section 4, please report the size bin structure used in the model (such as smallest bin size, bin spacing, and total number of bins), as well as the embedded radius bin structure for the new model (such as whether it is the same for all particle size bins), and how these were chosen.

1c. D* and k* that appear in equation 2 are not called out in the text. Please clarify.

1d. Please add at least one example of shell molar ratios, including nucleation rates as a function of shell and size bin. The non-monotonic behaviors seen in Figure 5 and Appendix B leave me skeptical about whether the model is numerically converged, and this would help another group reproduce your results, in addition to aiding some explanations.

2. Statement of the problem

2a. In the introduction, please be more quantitative about what is meant by "low temperature" (page 2, line 5).

2b. What is the significance of the clouds being sub visible and how is that defined?

2c. Page 2, lines 6–7: Quantitatively by how much are numbers and supersaturations

lower and higher than expected, respectively? In other words, how much of a problem is this?

2d. Page 2, line 14: What is meant by the "magnitude" of "both the physical and chemical aerosol properties" in the TTL?

2e. Page 2, line 14: Does the reported composition differ in any known way from aerosol particles elsewhere in the tropical upper troposphere? Please provide context.

2f. Page 2, line 20: Are you referring specifically to the tropical "upper" troposphere? Please provide some quantitative or definitional indication of what zone you're referring to, and how it may be distinguished from cirrus globally. I would also replace "were" with "are".

2g. Page 2, line 28: Grammatically, 'however' should be 'but' or a new sentence begun. That said, I don't understand how these clauses are oppositional, so I would just start the second clause as a new sentence with "Further..." Am I missing something?

3. Explanation and interpretation of results

3a. Page 11, line 1: This statement is not true for aerosol size distribution 2 as far as I can tell. Please note and explain that to the reader.

3b. page 11, line 8: I think the authors have the model output to evaluate this hypothesis without extensive additional calculations. Is there a reason that they stopped at suggesting it?

3c. Bottom of page 12 and Figure 6: I do not understand how the control model can give the same result if freezing is occurring at different temperatures and pressures along the trajectory owing to different initial supersaturation. Freezing rates should depend on temperature and pressure trajectory, right? What am I missing?

3d. Figure 6: Please report in the text which aerosol size distribution is used for these simulations.

3e. Also Figure 6: Does the trend change if the aerosol used changes or the cloud base temperature changes? As a reader, I can't place this monotonic trend into the context of obviously non-monotonic trends discussed thus far.

3f. Bottom of page 13: Inside-out nucleation is based on the heterogeneous ice nucleus touching the air-water interface. Are you suggesting that the more hydrated core is touching the solution-air interface? That seems unlikely to me. What is keeping it less hydrated is the proximity to the surface itself. Otherwise I don't see how this can be appropriately called inside-out nucleation, whether or not other authors have done so.

3g. Page 14, line 2: Isn't this just a matter of changing the initial or boundary conditions on your simulations? That does not strike me as complicated model development. Or am I missing something? Please explain whether any equations used to integrate the model need to be changed to account for this, and if so, why.

4. Significance of results relative to stated problem

4a. Last sentence on page 15 comparing simulated to grossly observed concentrations and concluding that the new model is better: I think this is quite an overreaching statement. Both models seem to span the enormous, five orders of magnitude range in ice crystal number concentrations reported equally well to me. Or perhaps neither model can reproduce the lowest ice crystal concentrations? And isn't the control simulation the only one that is generating the highest ice crystal number concentrations measured? I think the main point is just to show how the models differ and that both models reproduce values within the (huge) observed range, but the comparison with observations at this level is not distinguishing between these models meaningfully.

4b. Please discuss quantitatively whether differences in terminal supersaturation and ice number concentration in control versus new models are consistent with the model deficiencies discussed in the introduction, which have motivated this work. Quantitatively, specifically, what degree of ice supersaturation and number concentration respective excesses and deficiencies relative to expectations from control-type models

have been observed and are they roughly similar to what you are finding in your new versus old model as a function of temperature (which look to be an order of magnitude in number concentration and 10% in terminal supersaturation at the upper end)?

5. Minor suggestions

5a. Can Table A2 be placed after Table 1? I see no need to put so little material that is actually very relevant into an appendix. Or put the figure in the appendix and retain the table, which is more quantitatively understandable vis-a-vis past data sets for those experienced with aerosol measurements (i.e., one can read off a meaningful number concentration). Can the columns be labeled size distribution 1, 2 and 3 or made to otherwise match the column headings in Figure 5? References can be lowered to table line item. I had to keep flipping back and forth between appendix and text with figures to remember the number concentrations and basic differences.

5b. Figure 5: Rather than writing the enhancement and suppression regions, could the authors add some contours that show where the differences are larger than, say, 10% and 50% or similar relevant round numbers?

---

## Author Comment (AC1) · 10 Sep 2019

**Response to reviewer comments**

Kathryn Fowler[1], Paul Connolly[1], and David Topping[1]

[1]Centre for Atmospheric Science, The School of Earth and Environmental Sciences, The University of Manchester

**Correspondence:** Paul Connolly (paul.connolly@manchester.ac.uk)

We would like to thank the referees for their positive and constructive comments, which we have used to improve our manuscript. Attached is a PDF showing where changes have been made to the original version of the submitted manuscript and below are our specific responses to the referee comments.

**Response to anonymous referee #1**

My rating of the paper is, however, 'major revisions' for the reason I will explain in the following: unfortunately, the atmospheric conditions framing the paper do not match the presented results. The authors claim to help explain observations of very few ice crystals and high supersaturations in the very cold (< 205 K) tropical tropopause layer (TTL). Observed conditions in TTL cirrus clouds are: low ice particle concentrations around 0.005 - 0.1 cm-3, vertical velocities most frequently around a few cm/s (or even slower), infrequent waves up to 2 m/s (the low/high ice concentrations are found in the slow/fast updrafts). The conditions where suppression of homogeneous ice nucleation is found in the paper (Figure 8) are: ice particle concentrations > 3 cm-3 at vertical velocities 0.1 – 1 m/s. These vertical velocities and thus ice concentrations represent the atmospheric range of gravity waves, e.g. behind mountains, or convection (see for example Kärcher and Lohmann, 2002), but can not be extrapolated to TTL conditions. The results shown in Figure 8 clearly indicate that for TTL conditions the new and control simulations do not greatly differ, that means that no further understanding of the TTL ice concentrations can be obtained from this study.

The motivation for our study was that ultra viscous aerosol could affect the rate of ice nucleation at the low temperatures found in the TTL (Zobrist et al., 2008; Murray, 2008; Murray and Bertram, 2008; Froyd et al., 2010). However, we tested the models over a wide range of initial conditions to consider where the models were most sensitive to aerosol viscosity. Our findings show that aerosol viscosity was not important at the low up-draft speeds of the TTL, instead a difference is observed at vertical up-drafts corresponding to gravity waves. Thank you for drawing our attention to the flow of reasoning in our paper and the lack of link between our motivation and conclusions. We have therefore rephrased the motivation and stated that our model is tested over a greater range of initial conditions to better understand how composition dependant aerosol viscosity affects ice nucleation under a range of atmospheric conditions.

**Title:** From the title I did not have a good idea of the content of the paper. I would suggest to replace 'supercooled water' by 'ultra viscous particles', because the term 'supercooled water' to me implies homogeneous freezing of liquid cloud

drops at -38C.

We agree that this change means the title better represents the content of the paper.

**p 2, l 7-9:** 'These observations suggest that our current understanding of the ice formation mechanisms and therefore methods of modelling the formation of low temperature cirrus clouds are incorrect or incomplete (Peter et al., 2006; Krämer et al., 2009 ; Jensen et al., 2010).' Many more publications treating the topic appeared later - in case you mention the TTL in the next version of the manuscript, the most important newer studies should also be cited.

This has been addressed by including the references of Krämer et al. (2016) and Lee et al. (2018), which refer to the continued difficulties in understanding low temperature ice nucleation and the possible explanation of convective up-draft into the TTL.

**p 6, l 20:** '... extremely high number concentrations of organic aerosol have been observed in the upper troposphere near to regions of deep convective outflow in the tropics (Andreae et al.,2018).' The observations of Andreae et al. (2018) are well below the TTL, so is this statement really relevant for the study?

As suggested, we have reframed the study, as we were first motivated by the existence of viscous aerosol in the TTL, but tested the two parcel models over a wide range of atmospheric conditions to better understand the implications of viscous aerosol on homogeneous ice nucleation. Therefore we use this reference to show that the models were initiated under laboratory conditions were also atmospherically relevant.

**P 10, l 5:** '... up-draft velocities typically found in the tropical tropopause layer from Table 1.' The updraft range used for the study is not typical for the TTL, in particular the large scale updraft is lower, see comments to Table 1.

Agreed and amended.

**Figures 4 and 5:** The simulations shown in Figure 4 are (as can be seen from Figure 5) those with almost the largest effect on ice nucleation (Sze distribution 3, 0.6 m/s). The interpretation of the results regarding TTL cirrus are based mainly on this scenario. However, these conditions are untypical for the TTL - especially the updrafts are much slower (see comment on Table 1). For the TTL, the scenario shown at the bottom of the left column of Figure 5 would be most appropriate, though even here the updraft is rather high (only a few cm/s is typical in the TTL).

Agreed that our Figure 5 shows the highest limit of TTL updraft conditions in the lower left panel. Therefore we have clarified this by stating: "Our findings suggest that viscous organic aerosol, such as $\alpha$-pinene SOA, are expected to have no effect on the rate of homogeneous ice nucleation even at the highest limit of up-draft velocities found in the tropical tropopause layer as shown in the lower left panel of Figure 5 (size distribution 1, $0.1 \mathrm{ms}^{-1}$).".

**Figure 8:** This Figure shows that the suppression of homogeneous ice nucleation occurs for ice particle concentrations larger than 3 cm-3. The low concentrations that are under discussion, however, are in the range of 0.005 – 0.1 cm-3 (Krämer et al, 2009; Jensen et al., 2013; Spichtinger and Krämer, 2013), where no difference between the control and the new cloud parcel model is visible, or this range is not covered by the simulations.

The motivation for the study was that viscous aerosol in the upper troposphere could be a possible explanation for the

observed ice concentrations. However, after developing the new bin diffusion model we are able to say that viscosity probably does not account for the differences between observations and the standard homogeneous ice nucleation theory.

**Table 1:** The parameter spaces for TTL cirrus clouds specified in this table are partly not correct: Temperature: 185 – 205 K, Updraft: 0.01 – 2 m/s, or even slower in the TTL, Spichtinger and Krämer (2013), ACP, also Jensen et al. (2012), JGR.
We agree that this is confusing, so we now address the table as the parameter space for model testing rather than conditions typical of the TTL.

**Response to anonymous referee #2**

1. Reproducibility

    (a) Is the ascent adiabatic with respect to the model parcel? If so, parcel temperature must account for the latent heat of condensation. Please explain how environmental temperature and water vapor mixing ratio are calculated during parcel ascent.
    The accent is adiabatic (this has been clarified in the model description) and an accompanying model development paper with a more detailed model description will be released with the open source code to replicate the model runs.

    (b) Somewhere before section 4, please report the size bin structure used in the model (such as smallest bin size, bin spacing, and total number of bins), as well as the embedded radius bin structure for the new model (such as whether it is the same for all particle size bins), and how these were chosen.
    Agreed. See the paragraph accompanying Figure 2.

    (c) D* and k* that appear in equation 2 are not called out in the text. Please clarify.
    Amended.

    (d) Please add at least one example of shell molar ratios, including nucleation rates as a function of shell and size bin. The non-monotonic behaviors seen in Figure 5 and Appendix B leave me skeptical about whether the model is numerically converged, and this would help another group reproduce your results, in addition to aiding some explanations.
    We are finishing off an accompanying model development paper, which goes into detail about the model workings along with open source code. We believe that this paper will support those groups that would like to replicate our results. Through the development of our cloud parcel model, convergence was the main issue we had due to the large range in diffusion timescales and how this fitted in with the rising cloud parcel. We found that our solution converged once moving the diffusion solver into the cloud parcel ode solver, in doing this the model was able to use small enough time steps to ensure a smooth converging solution.

2. Statement of the problem

(a) In the introduction, please be more quantitative about what is meant by "low temperature" (page 2, line 5).

Amended. By low temperature we mean <205K.

(b) What is the significance of the clouds being sub visible and how is that defined?

Amended. Subvisible clouds have a low optical depth of less than 0.03 and are associated with very low ice crystal number concentrations.

(c) Page 2, lines 6–7: Quantitatively by how much are numbers and supersaturations lower and higher than expected, respectively? In other words, how much of a problem is this?

Measurements of ice number concentrations in the TTL of between 0.005 and 0.1 cm$^{-3}$ (Krämer et al., 2009; Jensen et al., 2013; Spichtinger and Krämer, 2013), however homogeneous theory (as shown in figure 8) predicts between 0.05 and 1 cm$^{-3}$ for the low temperatures and updraft speeds in the TTL. We have amended the introduction to state that its an order of magnitude less than expected.

(d) Page 2, line 14: What is meant by the "magnitude" of "both the physical and chemical aerosol properties" in the TTL?

Magnitude refers to the size of the TTL. We have reworded the sentence for clarity.

(e) Page 2, line 14: Does the reported composition differ in any known way from aerosol particles elsewhere in the tropical upper troposphere? Please provide context.

For this study we are more interested in the phase of aerosol particles in the atmosphere and have referred to papers by Shiraiwa et al. (2017) and Maclean et al. (2017) for this later in the introduction.

(f) Page 2, line 20: Are you referring specifically to the tropical "upper" troposphere? Please provide some quantitative or definitional indication of what zone you're referring to, and how it may be distinguished from cirrus globally. I would also replace "were" with "are".

The motivation for our study was the tropical tropopause layer, where there have been inconsistent measurements of ice number concentrations, however the study by Shiraiwa et al. (2017) is a global aerosol composition study and therefore refers more generally to the upper troposphere. We have clarified this point.

(g) Page 2, line 28: Grammatically, 'however' should be 'but' or a new sentence begun. That said, I don't understand how these clauses are oppositional, so I would just start the second clause as a new sentence with "Further. . ." Am I missing something?

Agreed, this reads much better.

3. Explanation and interpretation of results

(a) Page 11, line 1: This statement is not true for aerosol size distribution 2 as far as I can tell. Please note and explain that to the reader.

Clarity added.

(b) page 11, line 8: I think the authors have the model output to evaluate this hypothesis without extensive additional calculations. Is there a reason that they stopped at suggesting it?

The reader is advised to refer to figure 4 to support this statement.

(c) Bottom of page 12 and Figure 6: I do not understand how the control model can give the same result if freezing is occurring at different temperatures and pressures along the trajectory owing to different initial supersaturation. Freezing rates should depend on temperature and pressure trajectory, right? What am I missing?

Model simulations in Figure 6 have been initiated along the same pressure and temperature trajectory and aerosol particles initiated in equilibrium with these conditions. You are correct, in the control model we notice that all simulations initiated along the trajectory follow the black line. However, diffusion is composition dependant and therefore affects the rate at which the particles equilibrate and the number of frozen particles.

(d) Figure 6: Please report in the text which aerosol size distribution is used for these simulations.

Thank you for pointing this out, we have added this to the figure caption.

(e) Also Figure 6: Does the trend change if the aerosol used changes or the cloud base temperature changes? As a reader, I can't place this monotonic trend into the context of obviously non-monotonic trends discussed thus far.

This trend does change if cloud base temperature changes, which is what is shown in Figures 4 and 5. However, what we are showing here is that the history and processing of viscous aerosol particles needs to be accounted for and considered in further study.

(f) Bottom of page 13: Inside-out nucleation is based on the heterogeneous ice nucleus touching the air-water interface. Are you suggesting that the more hydrated core is touching the solution-air interface? That seems unlikely to me. What is keeping it less hydrated is the proximity to the surface itself. Otherwise I don't see how this can be appropriately called inside-out nucleation, whether or not other authors have done so.

We have removed this statement from the paper as it does not add anything to the study. I have not fully developed the idea and did not mean to include it into the final version.

(g) Page 14, line 2: Isn't this just a matter of changing the initial or boundary conditions on your simulations? That does not strike me as complicated model development. Or am I missing something? Please explain whether any equations used to integrate the model need to be changed to account for this, and if so, why.

Further model development would be required to trace frozen particles, and then solve for diffusion through an ice lattice while these particles are in a solid state. We have clarified the developments that are required in the text.

4. Significance of results relative to stated problem

(a) Last sentence on page 15 comparing simulated to grossly observed concentrations and concluding that the new model is better: I think this is quite an overreaching statement. Both models seem to span the enormous, five orders of magnitude range in ice crystal number concentrations reported equally well to me. Or perhaps neither model can reproduce the lowest ice crystal concentrations? And isn't the control simulation the only one that is generating the

highest ice crystal number concentrations measured? I think the main point is just to show how the models differ and that both models reproduce values within the (huge) observed range, but the comparison with observations at this level is not distinguishing between these models meaningfully.

Here we wanted to show that we had considered realistic atmospheric observations when analysing the data produced by our numerical models. However, we have not conveyed the reason for this inclusion well, which was to show that the models cover the magnitude in the range of ice crystal number concentrations. We have now clarified this in the text.

(b) Please discuss quantitatively whether differences in terminal supersaturation and ice number concentration in control versus new models are consistent with the model deficiencies discussed in the introduction, which have motivated this work. Quantitatively, specifically, what degree of ice supersaturation and number concentration respective excesses and deficiencies relative to expectations from control-type models have been observed and are they roughly similar to what you are finding in your new versus old model as a function of temperature (which look to be an order of magnitude in number concentration and 10% in terminal supersaturation at the upper end)?

As you have observed, the greatest differences between the control and new model simulations were around an order of magnitude in ice number concentration and 10% in terminal supersaturation. We initially intended to compare our model simulations with results from published cloud chamber studies as this would have allowed us to start model simulations with initial conditions corresponding to the controlled experimental conditions. However, the recent studies using $\alpha$-pinene in chamber experiments have published conflicting results, which vary from observations of heterogeneous ice nucleation processes, inhibited homogeneous ice nucleation and no differences observed from homogeneous ice nucleation (Möhler et al., 2008; Ladino et al., 2014; Wagner et al., 2017). Since current observations of controlled laboratory experiments of ice nucleation on $\alpha$-pinene aerosol do not agree, we decided that making a quantitative comparison was not possible at this stage. Therefore our main recommendation for further study is to be able to use the model alongside an experiment.

5. Minor suggestions

(a) Can Table A2 be placed after Table 1? I see no need to put so little material that is actually very relevant into an appendix. Or put the figure in the appendix and retain the table, which is more quantitatively understandable vis-a-vis past data sets for those experienced with aerosol measurements (i.e., one can read off a meaningful number concentration). Can the columns be labelled size distribution 1, 2 and 3 or made to otherwise match the column headings in Figure 5? References can be lowered to table line item. I had to keep flipping back and forth between appendix and text with figures to remember the number concentrations and basic differences.

Agreed and changed.

(b) Figure 5: Rather than writing the enhancement and suppression regions, could the authors add some contours that show where the differences are larger than, say, 10% and 50% or similar relevant round numbers?

We agree this conveys the value of the enhancement and suppression more clearly to the reader. Thank you for your comment.

**References**

[revised manuscript text omitted]